# Structure-from-motion photogrammetry demonstrates variability in coral growth within colonies and across habitats

**Ines D. Lange**[1]*, **Ana Molina-Hernández**[2], **Francisco Medellín-Maldonado**[2], **Chris T. Perry**[1], **Lorenzo Álvarez-Filip**[2]

**1** Geography, Faculty of Environment, Science and Economy, University of Exeter, Exeter, United Kingdom, **2** Biodiversity and Reef Conservation Laboratory, Unidad Académica de Sistemas Arrecifales, Instituto de Ciencias del Mar y Limnología, Universidad Nacional Autónoma de México, Puerto Morelos, Quintana Roo, México

* i.lange@exeter.ac.uk

**Data Availability Statement:** The raw data table is available from the University of Exeter Repository ORE at https://doi.org/10.24378/exe.4285.

## Abstract

Coral growth is an important metric of coral health and underpins reef-scale functional attributes such as structural complexity and calcium carbonate production. There persists, however, a paucity of growth data for most reef-building regions, especially for coral species whose skeletal architecture prevents the use of traditional methods such as coring and Alizarin staining. We used structure-from-motion photogrammetry to quantify a range of colony-scale growth metrics for six coral species in the Mexican Caribbean and present a newly developed workflow to measure colony volume change over time. Our results provide the first growth metrics for two species that are now major space occupiers on Caribbean reefs, *Agaricia agaricites* and *Agaricia tenuifolia*. We also document higher linear extension, volume increase and calcification rates within back reef compared to fore reef environments for four other common species: *Orbicella faveolata*, *Porites astreoides*, *Siderastrea siderea* and *Pseudodiploria strigosa*. Linear extension rates in our study were lower than those obtained via computed tomography (CT) scans of coral cores from the same sites, as the photogrammetry method averages growth in all dimensions, while the CT method depicts growth only along the main growth axis (upwards). The comparison of direct volume change versus potential volume increase calculated from linear extension emphasizes the importance of assessing whole colony growth to improve calcification estimates. The method presented here provides an approach that can generate accurate calcification estimates alongside a range of other whole-colony growth metrics in a non-invasive way.

## Introduction

In the Caribbean, a combination of disease, overfishing, pollution and climate change over the past decades has led to a region-wide decline in coral cover [1], reef complexity [2] and calcium carbonate budgets [3]. Critically, many reef communities are shifting towards assemblages dominated by slow-growing, non-framework-building taxa [4], a trend exacerbated by

**Funding:** This project was funded by the British Ecological Society (https://www.britishecologicalsociety.org/) through a research grant (SR21\100477) to ID Lange. The funders had no role in study design, data collection and analysis, decision to publish, or preparation of the manuscript.

**Competing interests:** The authors have declared that no competing interests exist.

the recent Stony Coral Tissue Loss Disease (SCTLD) outbreak [5]. The key metric to quantify the functional impacts of these changes is coral growth, underpinning reef health indices such as the *ReefBudget* method [6,7] and the *Reef Functional Index* [8]. A recent analysis of available growth data in the Caribbean reveals a major paucity of data for most coral species (76% of species with <3 publications on growth rates), but especially so for taxa that are becoming increasingly dominant [9]. One reason for this data deficiency is that traditional methods to determine coral growth rely either on x-radiography (X-ray) or computed tomography (CT) of coral cores, which only supports data acquisition for some massive coral species that display distinct annual growth bands, or on staining with Alizarin, which is risky and invasive as corals must be successfully stained, recovered, and killed to measure colony growth. Furthermore, coral growth rates tend to vary considerably in response to light, water quality, temperature, and aragonite saturation state [10], but studies on growth responses to environmental conditions and across different habitats only exist for a few species.

Coring and staining are both used to measure linear extension rates, which is the most common metric of coral growth and defined as unidirectional change in branch length or colony radius [10]. Linear extension may vary greatly among species and is typically much higher for branching or plating compared to massive and submassive growth forms [11]. Annual calcification (change in skeleton mass) can provide a more comparable measure of growth across morphological growth forms [10] but is more difficult to quantify *in situ* as it requires detachment of colonies for weighing or volume displacement. Alternatively, calcification may be calculated indirectly by multiplying linear extension rates or volume change with the bulk density of the coral skeleton.

Emerging approaches, using underwater photogrammetry and structure-from-motion (SfM) software to create three-dimensional (3D) models of coral colonies [12,13], are now helping to address many of the above analytical constraints. Recent studies explored colony-scale growth patterns by either measuring extension, surface area or volume repeatedly for the same colony [14,15] or by overlying models from subsequent years and directly quantifying the change in dimensions [16]. Besides being non-invasive and thus allowing repeated measurements of the same coral colony over many years, photogrammetry-based methods are providing novel ways to quantify a whole range of different colony-scale growth metrics and to analyse intra-colony variability in growth.

Here we apply and further develop the photogrammetry method introduced by Lange and Perry (2020) [16] to fill important gaps in available growth rate data for six common Caribbean coral species. By overlying 3D coral models, constructed from photographs taken of the same colony in subsequent years, we directly quantify changes in colony linear extension, surface area and volume. This approach enables us to integrate growth in a three-dimensional space whilst also considering partial mortality of colonies between survey dates. We compare and discuss a range of colony-scale growth metrics (average and maximum linear extension, area change, volume change and calcification) and analyse species-specific differences between a fore reef and back reef environment to understand how growth rates vary across environmental conditions, even on small spatial scales.

## Methods

### Species, study sites and photographs

In January 2020, 52 colonies of the species *Orbicella faveolata*, *Porites astreoides*, *Siderastrea siderea*, *Pseudodiploria strigosa*, *Agaricia agaricites* and *Agaricia tenuifolia* were tagged and photographed at two reef sites close to Puerto Morelos, Quintana Roo, Mexico (Fig 1). These corals are among the most common Caribbean species and were still present after the SCTLD

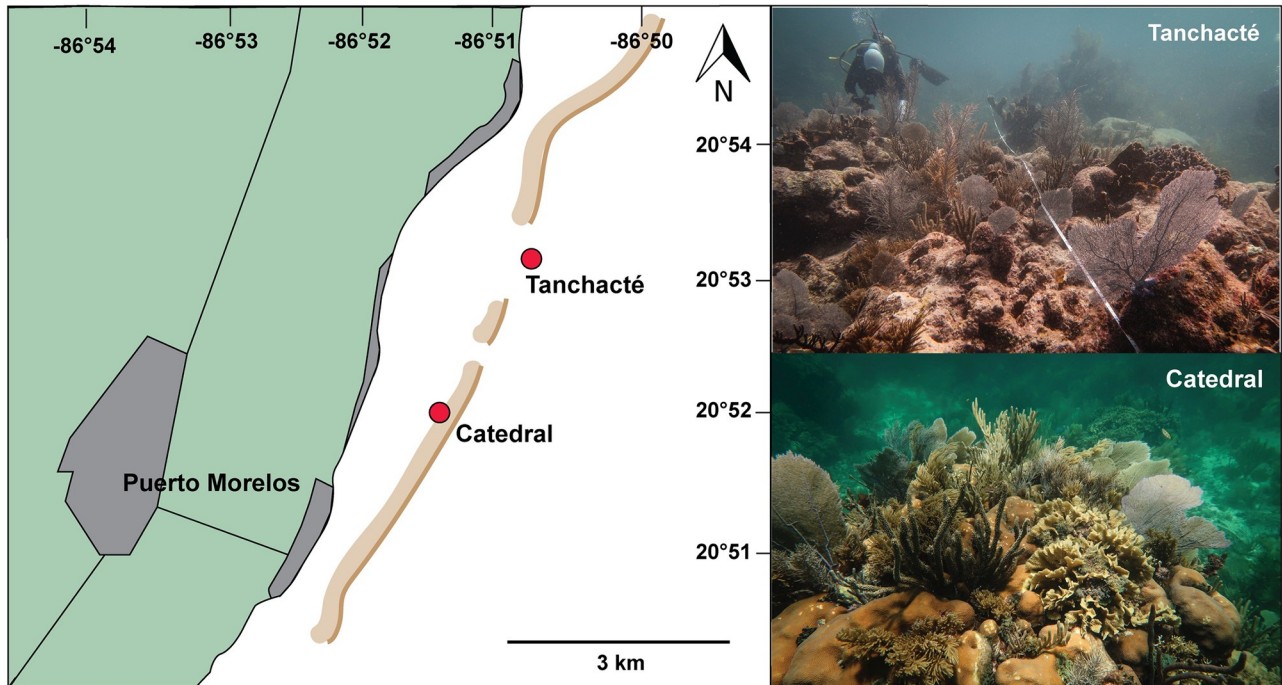

**Fig 1. Study sites close to Puerto Morelos, Mexico.** The reef crest as visible from satellite imagery is indicated in brown. Catedral is a shallow back reef environment (2 m depth) behind the reef crest and Tanchacté a deeper fore reef site (7 m depth) 2.5 km further north.

severely impacted reefs across the Mexican Barrier Reef [5]. The two surveyed reef sites represent two different habitats with distinct environmental conditions. The site Catedral (N20.86880˚ W86.85465˚) is a shallow back reef environment (2 m depth) close to the reef crest, while Tanchacté (N20.90711˚ W86.83224˚) is a more turbid but, due to its depth, less wave-exposed fore reef environment (7 m depth), approximately 2.5 km north of Catedral. Horizontal visibility was >10 m at Catedral and 7–8 m at Tanchacté during surveys, but is generally more variable at the fore reef and can be as low as 3–4 m. Sites were chosen for the ease of access and the occurrence of targeted coral species. The four massive coral species (*O. faveolata*, *P. astreoides*, *S. siderea*, *P. strigosa*) occurred at both sites, while submassive *A. agaricites* was only found at the fore reef site and foliose *A. tenuifolia* only at the back reef site. Sites were re-visited in March 2022 and while some colonies could not be re-located (n = 7) or were found dead (n = 11), we were able to replicate models for 34 colonies. Unfortunately, all tagged *A. tenuifolia* colonies experienced severe mortality between 2020 and 2022, but growth rate data for this species could be obtained by comparing 2020 models to models of the same colonies photographed in 2019 during a previous exploratory visit (n = 4), resulting in a total of 38 colonies for analysis.

Photographs were taken from multiple angles around each coral colony covering all aspects of the surface using a Canon Powershot G7X in an underwater housing (automatic underwater setting, no zoom, no flash) and a foldable ruler as a size reference [16]. Sea fans, soft corals and large fleshy macroalgae in the immediate surroundings of coral colonies were temporarily immobilised with a metal chain or diving weights to prevent excessive movement within a set of photographs, which inhibits the 3D modelling process.

## 3D modelling and measurement of growth metrics

Using Agisoft Metashape professional (v1.8.2), 3D models were constructed, scaled and exported as.ply following the workflow described in Lange and Perry (2020) [16]. Settings for each step are also provided as printable PDF tables in the Supplementary Material associated with this study (Table A in S1 Table). Dense point clouds of the same colony in subsequent years were then aligned in the software CloudCompare (v2.10.2) and isolated from the surrounding reef area by cutting around the colonies' peripheries. The alignment of point clouds was conducted manually using common features in the colony surrounding, which may affect distances between clouds. However, repeated alignment and measurements of the same colonies have resulted in very small deviation in linear growth (SD <1 mm) [16], and thus manual alignment represents a reliable and the best possible approach if fixed reference points cannot be deployed close to coral colonies due to conservation restrictions or time constraints. After models were aligned and isolated, the M3C2 plugin [17] was used to measure distances between models and calculate **average linear extension**. This tool quantifies distances between point *normals* in each cloud (i.e., points with the same growth orientation) and thereby increases the accuracy compared to the cloud-cloud distance tool, especially in areas of high local surface roughness [18]. Only for *A. tenuifolia*, average linear extension was estimated using the cloud-cloud distance tool, because the M3C2 plugin struggled to distinguish between fronds and to calculate vertical distances at narrow frond edges. Cloud-cloud distances were visualized for all colony models (Fig 2) and **maximum linear extension** was measured by either isolating the area of main growth (upper colony surface of *O. faveolata* and *S. siderea*) and averaging over all point distances, or by measuring distances at distinct points (n = 15) of maximum extension areas (bumps, ridges, columns and frond edges of *P. astreoides*, *P. strigosa*, *A. agaricites* and *A. tenuifolia*, respectively) using the 'Point picking' tool. Details and settings of this part of the workflow are provided in Table B in S1 Table.

Linear growth over the two-year study period was translated to annual extension rates by dividing the measured distances by the number of days between surveys and multiplying by 365.

We tested several approaches to calculate volume changes over time. First, both models were rotated together to display their main growth in vertical orientation, and distances between point clouds were estimated with the 2.5D volume tool in CloudCompare. This approach sums up distances between all points in z-direction (vertical), but only where the more recent model overlays the previous model, which is not always the case at the periphery. Therefore, the metric **vertical volume increase** may underestimate volume increases for more complex colony growth forms (e.g., *A. agaricites*) or colonies with extensive lateral growth (some colonies of *P. strigosa* and *P. astreoides*). For more accurate volume change estimates, dense point clouds were meshed in CloudCompare using the Poisson Surface Reconstruction plugin, which reconstructs a triangle mesh between points and creates a closed 3D model [19,20]. Meshes were filtered to only include the modelled colony surface, for which area was measured to calculate **area increase** between survey dates. Previous studies have then simply closed the hole at the bottom of the mesh to calculate the volume of each coral model individually, which works well for single *Acropora* branches with a small attachment point [15] but is not an option for colonies spreading irregularly over large substrate areas. We thus developed a new workflow to directly measure **volume increase**, details of which are provided in Table C in S1 Table and illustrated in a tutorial video (S1 Video). In short, the two aligned surface meshes were imported to the program Meshmixer (Autodesk, v3.5.474), a free software for 3D printing. After closing small holes, smoothing the boundaries, and flipping the *normals* of the 2020 mesh, the boundaries of meshes were fused to create a closed and measurable volume

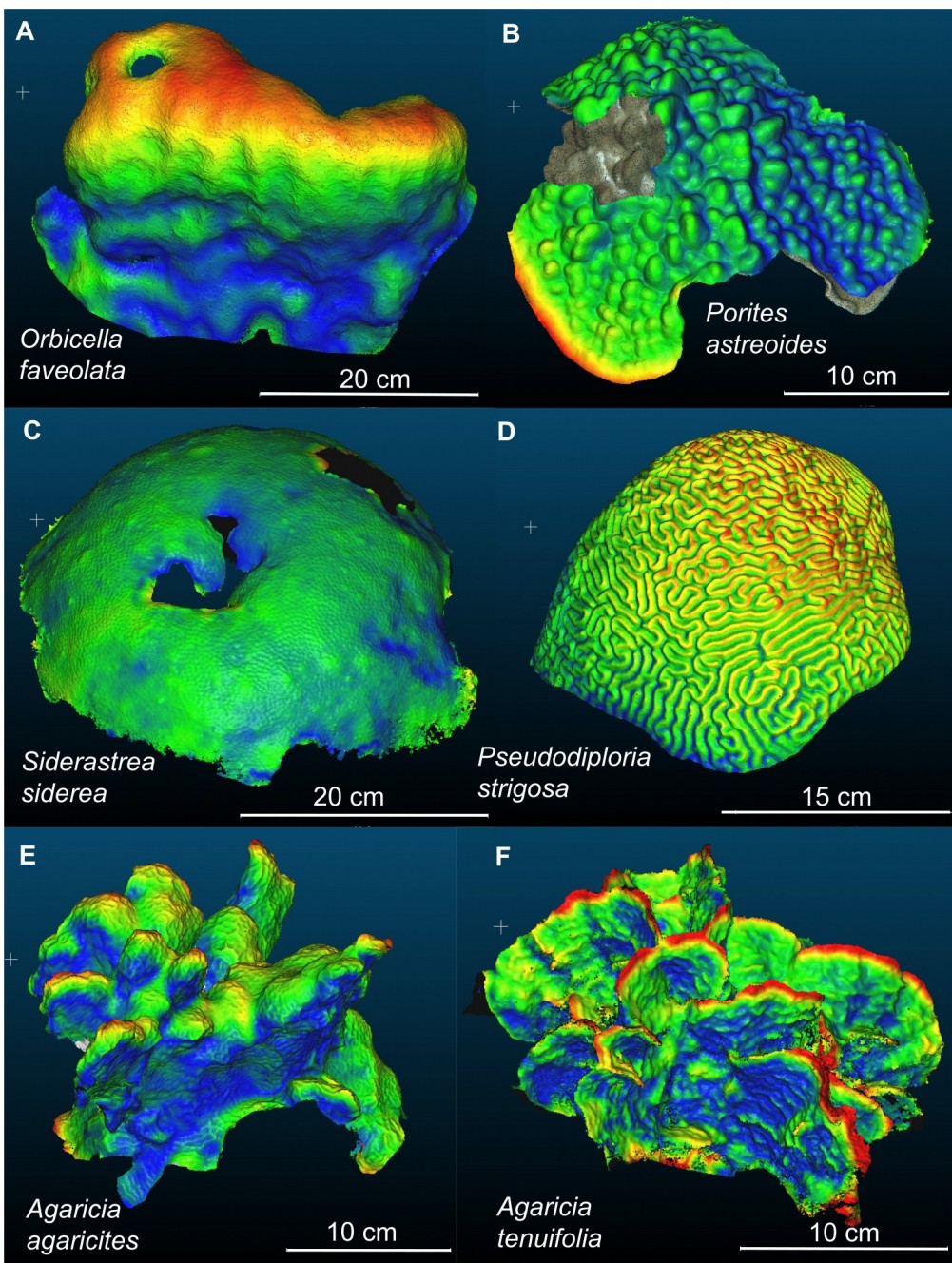

**Fig 2. Colony-scale growth of six Caribbean coral species.** The same coral colonies were photographed in subsequent years and colours display the distance between dense point clouds (red > yellow > green > blue) overlaid in the software CloudCompare. Note that highest growth often occurs on upper colony surfaces (A, C, D), around edges (B) or on bumps, columns and frond edges (B, E, F). A) The cavity in the top left area indicates the position of a core taken in 2021 for analysis with CT scans. B) Partial mortality between surveys shows the prior colony model below the more recent one.

between the two colony surfaces. While this approach worked very well for most colonies, structurally complex growth or lower point densities around colony boundaries sometimes caused issues with the model fusion, which increased time spent on manually fixing mesh boundary issues (loops, etc.) or joining the colonies manually using the Bridge tool and the Inspector tool to fill holes between bridges. For one fragmented *A. agaricites* colony, models of different colony parts were processed separately, and volumes were added. For *A. tenuifolia*, direct volume measurement between whole colonies was not feasible because the low resolution of point clouds in the cryptic area between foliose fronds (especially for 2019 models which were photographed with a lower resolution) caused high uncertainties and large gaps in surface meshes. To counteract this issue, we chose one frond from each colony that was well represented in both years. The frond was then isolated from the rest of the colony model, meshed, and its volume calculated separately for each year after closing the bottom hole in Meshmixer using the Bridge and Inspector tool. Volume increase was then calculated by subtracting 2019 volume from 2020 volume.

Annual rates for each of these growth metrics were calculated by dividing the area or volume increase by the number of days between surveys and multiplying by 365 before normalizing the result to the initial surface area of the colony.

To estimate annual **calcification rates** of coral colonies, volume increase was multiplied by species-specific skeletal bulk densities determined by optical densitometry analysis from CT scans of coral cores or fragments sampled from Catedral and Tanchacté back reef and fore reef sites in 2019 (Medellín-Maldonado et al., in preparation) and then normalized to year and initial surface area. Using species- and location-specific skeletal densities is important, as this parameter also varies among species and as a function of environmental conditions [10,21]. We additionally assessed **linear calcification** and **maximum linear calcification** by multiplying initial colony surface area with average or maximum linear extension rate and skeletal density. This approach is commonly used to estimate calcification from growth and density data obtained via X-ray analysis or Alizarin staining [10]. It is also applied in the census-based *ReefBudget* method to calculate reef-scale carbonate production from benthic survey data [7]. For branching and columnar taxa, *ReefBudget* calculations use a conversion factor reflecting the fact that only a part of the colony (branch/column tips) grows at the published linear growth rate. Following this procedure for *A. tenuifolia*, the calcification rate was multiplied by a conversion factor of 0.5, as only the upper half of fronds is actively growing.

## Comparison of growth rates

Species-specific growth metrics were compared between fore reef and back reef sites using multiple unpaired Welch t-tests with 1% False Discovery Rate in Multiple comparisons (one test for each species). Additionally, linear extension rates were contrasted to rates obtained at Catedral fore and back reef sites using a traditional method, i.e., measuring distances between density bands in CT scans of coral cores (Medellín-Maldonado et al., in preparation) using Welch t-tests. The different volume increase and calcification metrics were plotted against each other using colony-scale data to explore the agreement between different calculation methods. Finally, the proportion of the variance of one metric explained by the other was calculated, yielding an $R^2$ for the fit of plotted data against the ideal prediction function (y = x).

## Results

### Species- and habitat-specific growth

The visualization of colony-scale growth demonstrates the different growth strategies among coral species, with foliose *A. tenuifolia* growing only along frond edges and submassive *A.*

*agaricites* showing maximum growth at the tips of column-like structures (Fig 2E and 2F). While massive *O. faveolata*, *S. siderea* and *P. strigosa* displayed uniform growth across the colony or elevated growth in upper colony regions, *P. astreoides* demonstrated extensive growth (and mortality) along the colony edges (Fig 1B) and should therefore be classified as an encrusting/submassive rather than true massive growth form. Supportively, the morphology of *P. astreoides* has been formerly described as massive with a lumpy structure, submassive with knobs, columns or wedges protruding from an encrusting base, or plate-like with protrusions or columnar growth forms from the plate.

Average and maximum linear extension rates at both reef sites were highest for the two *Agaricia* species, followed by *O. faveolata* (Fig 3A and 3B), while volume increase and

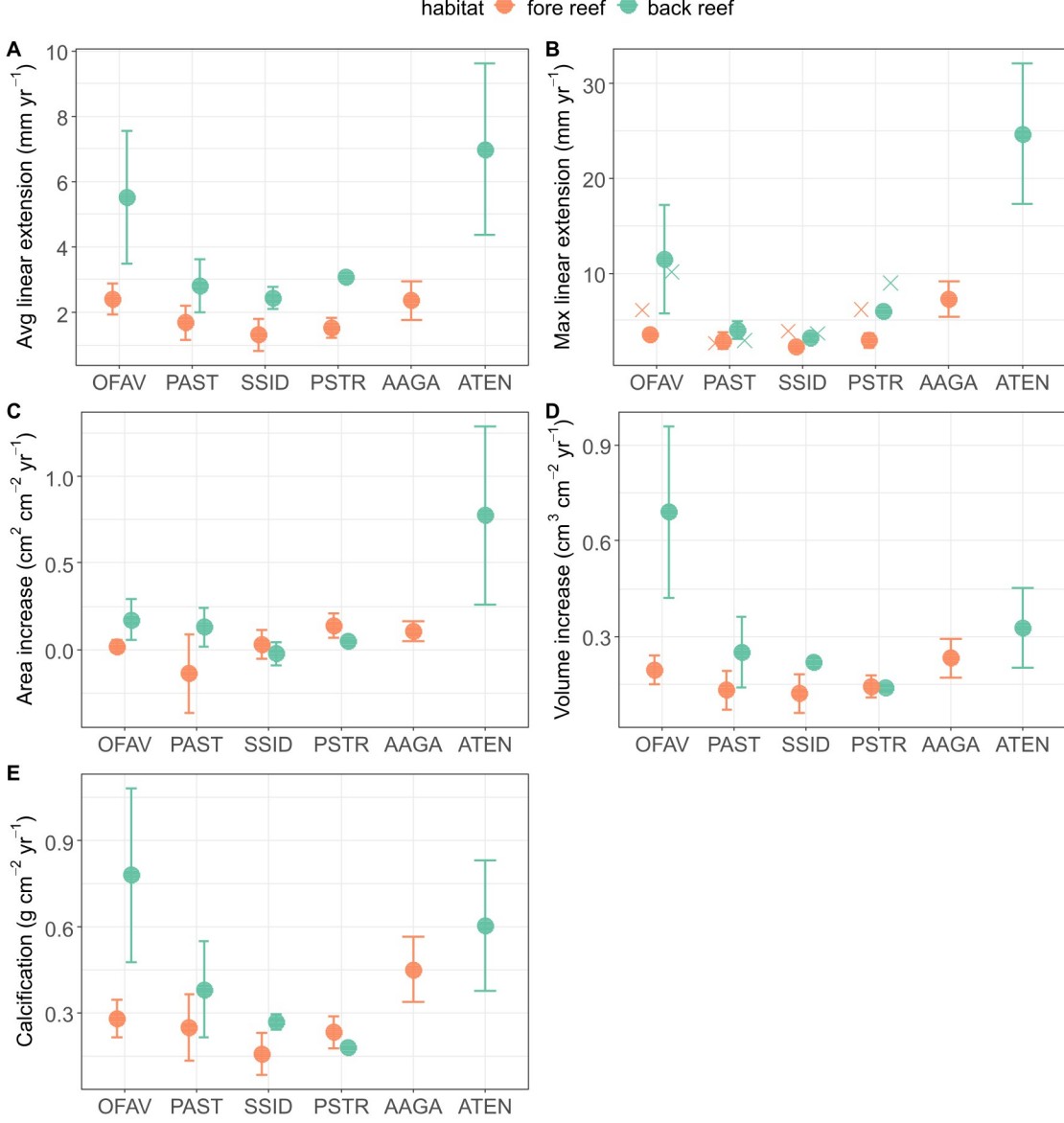

**Fig 3. Colony-scale growth metrics (average ± SD) for six Mexican Caribbean coral species comparing two reef sites.** Species: OFAV: *Orbicella faveolata*, PAST: *Porites astreoides*, SSID: *Siderastrea siderea*, PSTR: *Pseudodiploria strigosa*, AAGA: *Agaricia agaricites* and ATEN: *Agaricia tenuifolia*. Reef sites: Orange: Fore reef, green: Back reef. Panel B) additionally displays linear extension rates obtained by CT scanning of coral cores from the same sites (crosses; Medellín-Maldonado et al., in preparation).

**Table 1. Photogrammetry-derived colony-scale growth metrics (average ± SD) for six common Caribbean coral species at a fore reef site (7 m depth) and a back reef site (2 m depth) close to Puerto Morelos, Mexico.**

| species | habitat | n | avg linear extension (mm yr$^{-1}$) | max linear extension (mm yr$^{-1}$) | area increase (cm$^2$ cm$^{-2}$ yr$^{-1}$) | volume increase (cm$^3$ cm$^{-2}$ yr$^{-1}$) | calcification (g cm$^{-2}$ yr$^{-1}$) |
|---|---|---|---|---|---|---|---|
| *Orbicella faveolata* | fore | 3 | 2.41 ± 0.47 | 3.49 ± 0.43 | 0.02 ± 0.04 | 0.20 ± 0.05 | 0.28 ± 0.07 |
| | back | 3 | 5.54 ± 2.03 | 11.46 ± 5.75 | 0.17 ± 0.12 | 0.69 ± 0.27 | 0.78 ± 0.30 |
| *Porites astreoides* | fore | 4 | 1.69 ± 0.53 | 2.83 ± 0.86 | -0.14 ± 0.23 | 0.14 ± 0.06 | 0.25 ± 0.11 |
| | back | 4 | 2.81 ± 0.81 | 3.94 ± 0.92 | 0.13 ± 0.11 | 0.25 ± 0.11 | 0.38 ± 0.17 |
| *Siderastrea siderea* | fore | 4 | 1.31 ± 0.48 | 2.22 ± 0.40 | 0.03 ± 0.08 | 0.12 ± 0.06 | 0.16 ± 0.08 |
| | back | 3 | 2.45 ± 0.34 | 3.16 ± 0.39 | -0.02 ± 0.06 | 0.22 ± 0.02 | 0.27 ± 0.02 |
| *Pseudodiploria strigosa* | fore | 7 | 1.52 ± 0.31 | 2.92 ± 0.76 | 0.14 ± 0.07 | 0.15 ± 0.03 | 0.23 ± 0.05 |
| | back | 1 | 3.09 ± 0.00 | 5.93 ± 0.00 | 0.05 ± 0.00 | 0.14 ± 0.00 | 0.18 ± 0.00 |
| *Agaricia agaricites* | fore | 5 | 2.36 ± 0.59 | 7.25 ± 1.87 | 0.11 ± 0.06 | 0.23 ± 0.06 | 0.45 ± 0.11 |
| *Agaricia tenuifolia* | back | 4 | 6.99 ± 2.61 | 24.69 ± 7.42 | 0.78 ± 0.51 | 0.33 ± 0.13 | 0.61 ± 0.23 |

Metrics were derived by directly quantifying the change in dimensions between 3D coral colony models, except calcification which was calculated by multiplying volume increase with species- and habitat-specific skeletal densities. Abbreviations: avg—average; max—maximum.

calcification rates of *O. faveolata* surpassed the ones of *A. tenuifolia* at the back reef site (Fig 3D and 3E). For the four massive species, growth metrics differed more strongly across habitats than among species, with 2–5 times higher rates at the back reef site (Table 1), although differences were not statistically significant due to the low sample size (Welch t-tests p>0.5). Growth and calcification rates were especially small at the fore reef site for *S. siderea* and *P. strigosa*, and most *P. astreoides* colonies experienced area decrease over time due to substantial partial mortality between survey dates (Fig 3C).

## Comparison of different growth metrics

Maximum linear extension rates were 1.5 times (*P. astreoides*, *S. siderea*), 2 times (*O. faveolata*, *P. strigosa*) or 3 times (*A. agaricites*, *A. tenuifolia*) higher than the average rates (Fig 3A vs. Fig 3B) and compared well to growth rates obtained by CT scanning of coral cores from Catedral back reef and fore reef sites (Fig 3B). Area increase showed less intra- and interspecific variability compared to other growth metrics (Fig 3C) and despite the partial mortality of *P. astreoides* colonies at the back reef site, volume change was positive for all colonies. We therefore conclude that surface area increase is not a useful indicator of coral colony growth or calcium carbonate production over annual time scales, as this metric is highly sensitive to localised and partial mortality.

Volume increase and calcification rates showed species- and habitat-specific patterns very similar to linear extension rates (Fig 3D and 3E), although due to higher skeletal densities at the fore reef site we saw slightly less pronounced differences when comparing back reef and fore reef habitats (Fig 3E). The only exception was *P. strigosa*, which did not show higher volume increase and calcification at the back reef site despite higher linear extension rates. This was due to partial mortality of the coral colony between survey dates (and the low replication for this species and site).

The similarity in species- and habitat-specific patterns across growth metrics suggests that, at least for massive and submassive coral colonies, average linear extension rates are good indicators of colony-scale volume increase and annual calcification rates. For the foliose *A. tenuifolia*, volume increase was relatively low compared to its high linear extension rates, as growth is restricted to the narrow upwards facing frond edges.

## Volume and calcification estimates

Calcification rates can be estimated by multiplying volume increase or linear extension rates with skeletal densities, and we compared different approaches to quantify these metrics. Our results show that the 2.5D command in CloudCompare slightly underestimates actual volume increase, especially for larger and more complex colonies (Fig 4A), as not all the volume change is represented in the vertical growth direction. However, the fit to the ideal line of prediction of direct volume measurements (which we consider the most accurate volume metric) is very strong ($R^2$ = 0.92), indicating that this quick and easy tool for estimating volume increase is useful for massive colony growth forms. The fit to direct volume increase is even better when calculating volume changes from initial surface area and average linear extension rate (Fig 4B, $R^2$ = 0.96), which represents 'potential' volume increase over time as it ignores partial mortality between survey dates. This approach is commonly used to calculate coral calcification from linear extension rates.

Linear extension rates are traditionally derived from X-ray or CT analysis of coral cores from the upper colony surface (main growth direction), while our photogrammetry approach

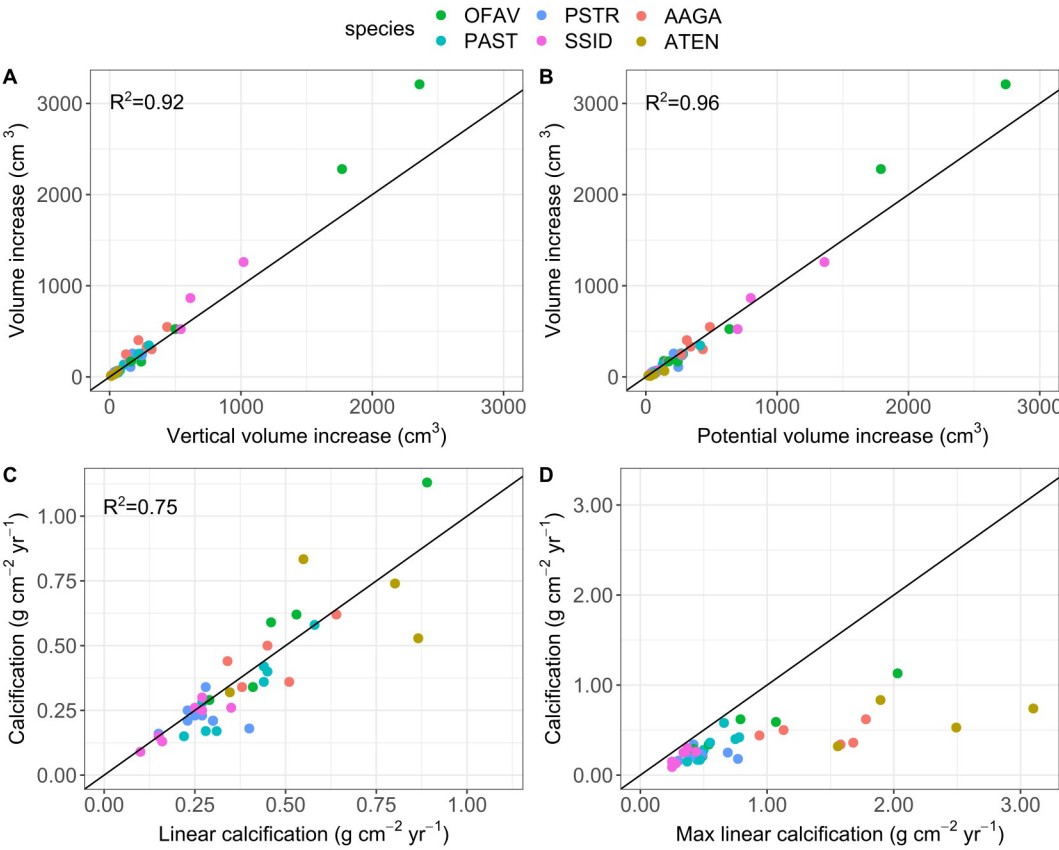

**Fig 4. Comparison of different methods to calculate volume increase and calcification rates.** Each point represents one colony (n = 38) of one of six species (OFAV: *Orbicella faveolata*, PAST: *Porites astreoides*, SSID: *Siderastrea siderea*, PSTR: *Pseudodiploria strigosa*, AAGA: *Agaricia agaricites* and ATEN: *Agaricia tenuifolia*). *Y-axes*: Best estimate volume increase (A and B) was measured directly between overlying surface meshes and multiplied by skeletal densities to yield best estimate calcification rates (C and D). *X-axes*: A) Vertical volume increase was calculated using the 2.5D volume command in CloudCompare. B) Initial surface area was multiplied with average linear extension to yield potential volume increase, and then C) multiplied by skeletal densities to yield linear calcification. D) Average linear extension was replaced by maximum linear extension rates to yield maximum linear calcification. $R^2$ in plots indicates the fit of data to the ideal line of prediction (y = x).

averages extension in all dimensions. To test the impact of growth direction on calcification estimates we multiplied the initial surface area of coral colonies with skeletal densities and either average or maximum linear extension rates, and then compared the results to calcification rates derived from direct volume measurements (which we argue is the most accurate metric). The spread of data is naturally wider than that for the comparison of different volume measurements, as the data range is smaller after being normalized to initial colony surface area. Using average linear extension rates slightly overestimates calcification for colonies experiencing partial mortality (mainly *P. astreoides*, Fig 4C), but the strong correlation to the ideal line of prediction ($R^2 = 0.75$ with *A. tenuifolia*, $R^2 = 0.81$ without) indicates that this simplified approach is a fairly accurate method to calculate calcium carbonate production rates from colony contour or area data, at least for massive and submassive growth forms. This however is only true if average linear extension rates across all growth directions are used for calculations, while using maximum linear extension rates at the top surface of colonies results in significant overestimation of calcification for all coral colonies (Fig 4D). The deviation was particularly high for the foliose species *A. tenuifolia* and the submassive species *A. agaricites*, which due to their morphological growth patterns display large differences between average and maximum linear extension.

## Discussion

In this study we provide a range of colony-scale growth metrics for six common Caribbean coral species, for some of which no, or no local, growth rates were previously available. Our data demonstrates varying growth strategies among species and different growth rates across habitats. A newly developed workflow provided with this study describes how to directly measure annual volume increase of coral colonies, supporting accurate calculation of colony-scale calcification rates for a range of species and growth morphologies.

### Species- and habitat-specific growth

Using SfM photogrammetry, average linear extension rates in the range of 1.3–7.0 mm yr$^{-1}$ were measured. Studies using X-ray analysis conducted in the same region and at similar depth (2–8 m) displayed slightly higher yet comparable linear extension rates for *O. faveolata* (range: 5.5–8.4 mm yr$^{-1}$) [22,23], *P. astreoides* (avg. ± SD: 4.3. ± 1.2 mm yr$^{-1}$) [24], and *S. siderea* (3.3 mm yr$^{-1}$ at fore reef, and 4.5 mm yr$^{-1}$ at back reef site) [25]. Growth studies in Barbados, Jamaica and St. Croix yielded comparable rates for *A. agaricites* (avg. ± SD: 2.8 ± 1.7, range: 1.1–4.8 mm yr$^{-1}$) [26–28]. The comparison supports the reliability of the photogrammetry approach to quantify linear extension rates of coral colonies in the mm to cm range, while avoiding the need for unreliable staining and destructive colony sampling or core recovery for X-ray/CT analysis. Critically, this new approach additionally allows an assessment of overall colony-scale growth and illustrates focal points of maximum extension that clearly vary with colony morphology.

The shape and size of coral colonies are known to determine how they interact with the environment and other species. Different growth strategies allow species to compete for space and resources, and directly influence demographic rates [29] and reef functions [30]. High linear extension rates of *A. tenuifolia* and *A. agaricites* and extensive expansion along the colony edges of *P. astreoides* suggest that the ability to rapidly occupy space has helped these weedy species to become dominant on today's (and likely tomorrow's) Caribbean reefs. As these species are small and do not notably contribute to reef structure or community calcification, the functional integrity of coral reefs in the region continues to decline [5,8,30]. The effects of shifting community composition on reef functions may however vary on small spatial scales,

as we show that growth rates clearly differ between closely spaced habitats, probably as a function of locally variable environmental conditions.

For all four species that occurred at both reef sites, linear extension and calcification rates were higher at the back reef compared to the fore reef site. Lower growth rates at the fore reef are potentially related to lower light levels at deeper depths [31], although prior studies did not find light effects on coral growth comparing 2 m and 7 m depth [23,28]. However, higher turbidity at the fore reef site possibly decreased light levels even further and/or lead to accumulation of fine sediments on colony surfaces, reducing photosynthesis of coral symbionts and feeding efficiency in corals [32]. Although we lack the necessary environmental data to explore the reasons for this pronounced coral growth response, our results emphasize the strong influence of environmental conditions and the high variability in growth even across small spatial scales. The same spatial pattern was visible in linear extension rates obtained from CT scans of coral cores at Catedral fore reef and back reef sites (Medellín-Maldonado et al., in preparation), demonstrating that our observations are clearly not a methodological artefact.

Interestingly however, CT scans showed up to 3-fold higher annual growth compared to photogrammetry-derived average linear extension rates. This may suggest a decrease in growth rates in recent years, possibly caused by chronic or short-term high temperature stress [33,34] or land-based anthropogenic stressors [35]. Additionally, three hurricanes in 2020 that caused dislodgment of coral colonies and rubble in the backreef environment could have possibly affected growth and partial mortality of some photographed colonies. Indeed, reduced growth rates for 2019 and 2020 were observed in several cores analysed with CT scans (Medellín-Maldonado et al., in preparation), while portrayed averages were calculated over 5–6 years prior to sampling (reflecting growth between 2014–2020) and may therefore be higher than our photogrammetry-derived rates (growth between 2020–2022).

However, a core taken from a photographed *O. faveolata* colony (Fig 2A) showed similar CT-derived annual linear extension rates between 2016 and 2020 (avg. ± SD: 9.6 ± 1.3 mm yr$^{-1}$) compared to the photogrammetry-derived rate for the upper part of the colony between 2020 and 2022 (10.6 ± 4.8 mm yr$^{-1}$), indicating that growth rates are comparable if samples are taken from the same colony region. Differences between the studies are therefore much more likely caused by methodological differences and inter-colony variability in growth, as cores are usually collected from the top of coral colonies [10] where extension rates are highest (e.g., 16% higher extension rates in vertical compared to horizontal cores for *Porites* sp. [36]). Indeed, our 3D models illustrate that many massive colonies showed elevated growth in upper colony regions (Fig 2) and CT derived rates agree more closely with maximum linear extension rates obtained in this study (Fig 3B). This means that while the photogrammetry approach produces estimates of whole colony growth, the rates derived from X-radiography and CT scanning reflect maximum extension rates along the main growth axis. Depending on species and growth morphology this leads to considerable differences in linear extension rates.

Similarly, calcification rates in our study were smaller than rates calculated from X-ray analysis for *O. faveolata* (0.97 g cm$^{-2}$ yr$^{-1}$) and *P. astreoides* (range: 0.71–0.80 g cm$^{-2}$ yr$^{-1}$) in the same geographic region [24,33], again reflecting the different growth directions used for calcification calculations, as coral cores in the referenced studies were taken from the upper colony surface.

## Photogrammetry to quantify coral growth

We demonstrate that structure-from-motion photogrammetry provides an accurate, non-invasive method to measure linear extension rates, with the caveat that there is no temporal resolution beyond the time of study as is possible with X-ray or CT analysis of coral cores [10].

While destructive coring often results in a very small sample size, the photogrammetry approach potentially allows the scaling of sample size more easily and cheaply than other methods. The most important advantages over traditional methods are however that photogrammetry allows to quantify a range of coral growth metrics, to compare growth in different colony parts and to average linear extension and volume increases across entire coral colonies. It also allows monitoring changes at the reef scale [12] and could be used to quantify colony-scale or population-scale mortality.

The newly developed workflow provided with this study describes how to directly measure annual volume increase in coral colonies, even when the colony base covers a large substrate area. While the manual alignment of point clouds introduces some uncertainty that may affect growth rates, repeated measurement of the same colonies have resulted in very low variation in linear growth (SD <1 mm) [16], which is much smaller than within-colony variation in growth. The alternative of measuring colony diameter, surface area or volume of each colony following a morphometric approach would introduce a much larger error, as the base of the colony is often difficult to define. Additionally, the direct comparison of models takes into account partial mortality between survey dates and integrates areas and seasons of slower or faster growth, and therefore allows realistic estimates of annual colony-scale calcification rates for a range of species and growth morphologies. It should be noted however that uncertainties for foliose corals such as *A. tenuifolia* are higher than for other species, as visual occlusion of lower colony regions complicates the construction of accurate surface meshes for the entire colony and volume increases in this study were therefore based on one frond per colony. Similar occlusion issues would arise when constructing models for densely branching (e.g., some *Acropora* spp., *Seriatopora* spp.) or whorl-shaped (e.g., *Turbinaria* spp.) growth forms. Enhanced illumination (flashlight/torch) and taking a higher number of high-resolution photographs would help to improve model building in these cases. However, despite the uncertainties associated with calculating volume increase for *A. tenuifolia* in our study, calcification data from Belize obtained by repeatedly weighing coral fragments yielded very similar rates (range: 0.45–0.68 g cm$^{-2}$ yr$^{-1}$ [37]).

Exploring different approached to estimate calcification rates, we showed that average linear extension rates multiplied with initial colony surface area and site-specific skeletal density measurements show a good fit with calcification rates calculated from direct volume measurements. This suggests that the simplified approach of using size measurements and linear extension rates is suitable for colony- or reef-scale carbonate production estimates. However, census-based methods such as the *ReefBudget* approach necessarily use available traditionally obtained linear extension data from coring and Alizarin staining, which often only consider the main growth axis i.e., resulting in what we would call maximum extension rates. As we show that using such rates can result in a large overestimation of calcification, the implications on colony- or reef-scale carbonate production estimates will have to be considered in future research. For the *ReefBudget* methodology this would suggest that improvements can be made by only using linear growth rates that consider growth in all directions, either from photogrammetry ([16], this study) or Alizarin staining studies that averaged extension over multiple directions [e.g., 11]. This data however remains scarce, as most traditional growth studies measured extension in only one direction [10]. Another idea may be to apply conversion factors, which are currently being used for columnar and branching growth forms, also for massive and submassive morphology taxa (i.e., applying maximum linear growth rates to 20–30% of the surface area and assuming the rest of the colony is growing at a lower rate). For foliose *A. tenuifolia*, the conversion factor of 0.5 worked reasonably well to relate average linear growth to calcification rates, whereas a conversion factor of 0.14 would be more appropriate if using maximum linear extension rates. For submassive *A. agaricites*, no conversion was necessary

using average linear rates, while maximum linear rates would require a conversion factor between 0.21 and 0.47. The large variability between *A. agaricites* colonies indicates that conversion factors do not work well for species with a very variable growth morphology and that it is more useful to use average extension rates in this case. Future work could usefully determine whether for massive colonies, general proportional values could be used for the area of colonies growing at maximum rates and the amount of growth reduction from top to sides, or if species-specific values would be necessary.

## Conclusion

Species- and site-specific growth rates are critical to determine important reef functions, and the different growth metrics for six common coral species we make available in this study provide a basis for improving assessments of functional indices [8] and reef carbonate production rates [7] in the Caribbean.

We demonstrate that structure-from-motion photogrammetry provides an accurate, non-invasive method to measure average linear extension over all dimensions and quantify additional key colony-scale growth metrics for a range of species and growth morphologies. Specifically, we provide a newly developed workflow describing how to directly measure annual volume increase to accurately constrain colony-level calcification rates.

The comparison of different calculation methods shows that annual calcification can also be quantified from average linear extension rates and skeletal densities. However, readily available growth rates from coring or staining studies mostly represent maximum extension rates along the main growth axis, and using such rates largely overestimate colony- and reef scale calcification. We therefore emphasise the need for generating more data on whole colony growth and routinely integrating measurements in different directions into all coral growth studies.

## Supporting information

**S1 Table. Workflows for 3D model construction and measurement of coral colony growth rates using Metashape, CloudCompare and Meshmixer.**
(PDF)

**S1 Video. Video tutorial guiding through the process of quantifying linear extension, maximum linear extension, area increase and volume increase of coral colonies using SfM photogrammetry.**
(MP4)

## Acknowledgments

We thank Esmeralda Pérez-Cervantes, Alba Gonzalez-Posada, Alexis Medina-Valmaseda and Nuria Estrada-Saldívar for assistance in the field and boat operators for logistical support.

## Author Contributions

**Conceptualization:** Ines D. Lange, Chris T. Perry, Lorenzo Álvarez-Filip.

**Formal analysis:** Ines D. Lange.

**Funding acquisition:** Ines D. Lange.

**Investigation:** Ines D. Lange, Ana Molina-Hernández, Francisco Medellín-Maldonado.

**Methodology:** Ines D. Lange.

**Visualization:** Ines D. Lange.

**Writing – original draft:** Ines D. Lange.

**Writing – review & editing:** Ana Molina-Hernández, Francisco Medellín-Maldonado, Chris T. Perry, Lorenzo Álvarez-Filip.

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
