## [Decision Letter · Decision Letter 0]

24 Oct 2022

PONE-D-22-25436Structure-from-motion photogrammetry demonstrates variability in coral growth within colonies and across habitatsPLOS ONE

Dear Dr. Lange,

Thank you for submitting your manuscript to PLOS ONE. After careful consideration, we feel that it has merit but does not fully meet PLOS ONE’s publication criteria as it currently stands. Therefore, we invite you to submit a revised version of the manuscript that addresses the points raised during the review process.

Authors should discuss the limits of their approach and how much these limits may affect their results. Reviewer 1 lamented the lack of fixed reference points. Commenting upon the potential error deriving from the procedure adopted is deemed necessary before acceptance.

We look forward to receiving your revised manuscript.

Kind regards,

Carlo Nike Bianchi

Academic Editor

PLOS ONE

**Journal Requirements:**

3. Please include a copy of Table 5 which you refer to in your text on page 5.

Reviewers' comments:

Reviewer's Responses to Questions

**Comments to the Author**

1. Is the manuscript technically sound, and do the data support the conclusions?

Reviewer #1: Partly

Reviewer #2: Yes

2. Has the statistical analysis been performed appropriately and rigorously? 

Reviewer #1: N/A

Reviewer #2: Yes

3. Have the authors made all data underlying the findings in their manuscript fully available?

Reviewer #1: Yes

Reviewer #2: Yes

4. Is the manuscript presented in an intelligible fashion and written in standard English?

Reviewer #1: No

Reviewer #2: Yes

5. Review Comments to the Author

Reviewer #1: The Authors presented a methodological approach for monitoring corals’ morphology changes over time using the structure from motion technology and open-source software to estimate the distance between point clouds of the same colony obtained from two monitoring events.

The corals scanning was carried out by positioning reference dimension items close to the subject. No fixed reference points were used during the monitoring to align point clouds generated from different monitoring events.

The alignment of two-point clouds was done manually and using a tool (S2 table, step 2 of the Sup. Materials). The error associated with this methodological step is not quantified or is not possible to be quantified, but it affect the measured coral growth rates.

Reviewer #2: The study has great interest and a high degree of novelty. It is very well thought out and resolved. Congratulations to the authors. I have only commented on some details (attached file) that could be considered in a new version of the manuscript.

6. PLOS authors have the option to publish the peer review history of their article (what does this mean?). If published, this will include your full peer review and any attached files.

Reviewer #1: No

Reviewer #2: No

---

## [Author Response · Author response to Decision Letter 0]

26 Oct 2022

see attached Response to reviewer document

---

## [Editor Report · Decision Letter 1]

31 Oct 2022

Structure-from-motion photogrammetry demonstrates variability in coral growth within colonies and across habitats

PONE-D-22-25436R1

Dear Dr. Lange,

We’re pleased to inform you that your manuscript has been judged scientifically suitable for publication and will be formally accepted for publication once it meets all outstanding technical requirements.

Kind regards,

Carlo Nike Bianchi

Academic Editor

PLOS ONE
---

## [Editor Report · Acceptance letter]

8 Nov 2022

PONE-D-22-25436R1 

Structure-from-motion photogrammetry demonstrates variability in coral growth within colonies and across habitats 

Dear Dr. Lange:

I'm pleased to inform you that your manuscript has been deemed suitable for publication in PLOS ONE. Congratulations! Your manuscript is now with our production department. 

Kind regards, 

on behalf of

Dr. Carlo Nike Bianchi 

Academic Editor

PLOS ONE